# From Extermination to Conservation: Historical Records of Shark Presence during the Early and Development Phase of the Greek Fishery

**DOI:** 10.3390/ani12243575

**Published:** 2022-12-17

**Authors:** Dimitrios K. Moutopoulos, Evridiki Lazari, George Katselis, Ioannis Giovos

**Affiliations:** 1Department of Fisheries & Aquaculture, University of Patras, 30200 Mesolongi, Greece; 2iSea, Environmental Organization for the Preservation of the Aquatic Ecosystems, 54645 Thessaloniki, Greece; 3Department of Biology, University of Padova, 35122 Padova, Italy

**Keywords:** cartilaginous fish, historical accounts, shark occurrence, marine environmental history

## Abstract

**Simple Summary:**

A thorough search of traditional and digital libraries was conducted for retrieving issues on the interaction of the Greek fisheries with shark species. A significant contribution was made through newspaper collections covering both the Athenian and provincial press including Crete Island for a century (1883–1983). Our historical search showed that large species were common in the Greek Seas, a phenomenon not suspected from datasets of modern fishery surveys. The detection of sharks due to local media gradually increased up to 1969, with most records being more frequent for the Aegean Sea, whereas the number of sharks being sighted declined leading up to the middle of 1980s. The crucial point is that a large number of these observations were related to shark attacks on people, whereas this is not currently evident. The historical records presented here show us that the knowledge of the past can motivate us to identify strategies and policies that can be more acceptable for communities and thus succeed for conservation.

**Abstract:**

The lack of historical data on shark presence, distribution, and status in the Eastern Mediterranean undermines efforts to manage and protect their populations. An exhaustive review of anecdotal references related to shark presence during the early and development phase of Greek fisheries (1883–1983) was conducted. In the early-20th century (1912), the first sighting of the presence of a dead shark was reported in the Ionian Sea. Later on, the presence of sharks gradually increased up to 1969, with most records being more frequent for the Aegean Sea, whereas the number of sharks being sighted declined leading up to the middle of 1980s. The increase in shark attacks during the mid-20th century led to a calling for culling of sharks in co-operation with the competent authorities promoting the permission to hunt sharks with firearms and offering rewards for killed individuals. A high number of these observations potentially resulted from shark attacks on people, whereas this is not currently evident. This is an indicator of the lower abundance of sharks in modern times and subsequently an alteration in the way that our current modern society is approaching the protection of such vulnerable species.

## 1. Introduction

Historical anecdotal data can improve our understanding of past system dynamics, enabling us to determine whether contemporary systems are acting within the historical range of variability exhibited before large-scale human impacts [1]. Historical information can also help us with the phenomenon of shifting environmental baselines that each generation subconsciously views as “natural” given the way the environment appeared in their youth [2]. The transition of knowledge from earlier periods, when human impacts were limited [3], could be a useful “repository” of knowledge aiming to re-evaluate management thresholds [4]. In this direction, the compilation of historical and “forgotten” science with modern natural observations has significantly increased nowadays, under the framework of Marine Historical Ecology and Marine Environmental History [5,6], incorporating a wide range of multi-disciplinary fields of science (e.g., [7]).

The Mediterranean Sea’s diverse biota provide a wide range of interactions with people and the local populations [8]. In this environment, sharks are among the oldest animals that are still alive today [9], and because of their biological traits (slow development, late maturity, and low fecundity), they are particularly vulnerable to overfishing [10]. In the present study, a thorough review of anecdotal references related to shark presence aims to evaluate shark–human interactions in the Greek waters during the early and developmental phase of the fisheries (1900-early 1980s). This effort will close the historical gap in knowledge on shark populations in the Eastern Mediterranean [11,12], and the data gathered could be used in the MEDLEM database, a repository of information on large elasmobranchs in the Mediterranean and Black seas [13]. It can also be used in assessments for the Red List by the International Union for Conservation of Nature (IUCN), which helps to understand the historical presence and distribution of species. Our results will also verify whether or not the reported shark attacks on people are well recorded in the International Shark Attack File (ISAF). This database was initiated in 1958, and more than 6500 individual investigations cover the period from the early 1500s to the present [14]. Evidence derived from the study of historical and archival documents, such as the one here, may be used for assessing the vulnerability of exploited species [15], including sharks [16]. Historical science may play an important role to comprehend present-day effects and conditions [17]. In this context, historical accounts on sharks may be extremely useful to add new data to their occurrence and distribution in poorly studied regions such as the Greek waters [18].

## 2. Materials and Methods

Greek fisheries officially started to getting organized in 1911 and, up to the early 1980s, these fisheries were shifting from an essentially pre-industrial stage toward the industrialization of fishing activities [19]. The present study dealt with research through Greek traditional and digital archives (i.e., newspapers, technical reports, and books) using the keyword “shark” during a century (1883–1983). Digital archives included the newspaper collection of the National Library of Greece (http://efimeris.nlg.gr/ns/main.html?fbclid=IwAR0n__4AKJQ-ci7BFEwxCxZmu-90qQRZhlGhyMSmmcpkvB9gThXnwQmwi8E (accessed on 10 September 2022)) consisting of the following journals: *Eleftheria*, *Empros*, *Macedonia*, *Rizospastis*, *Scrip*, and *Acropolis*. These archives cover both the Athenian and provincial press from 1883 and up to 1983, depending on the journal. Duplicate records of the same report published in different newspapers were excluded. For the newspapers in Crete Island, old archives were found in the online database (http://vikelaia-epapers.heraklion.gr/%CE%B5%CF%86%CE%B7%CE%BC%CE%B5%CF%81%CE%AF%CE%B4%CE%B5%CF%82/ (accessed on 10 September 2022)).

Elasmobranch records were also searched through the Mediterranean Elasmobranchs Citizen Observations (MECO) Project (www.facebook.com/pg/theMECOproject, accessed on 6 August 2022) [18], which launched in 2014 and aimed to collate knowledge on elasmobranch occurrence, seasonality, and distribution using citizen science and social media. The project involves the collaboration of local scientists who are searching the media, contacting the public, and creating a large, verified database of elasmobranch observations.

For the visualization of the reports, the open-access software QGIS ver 3.16.7-Hannover [20] was used. The percentage contribution of the reported sharks in Greek waters were estimated for overall and area disaggregated values across years.

## 3. Results

Overall, 197 historical records on shark presence in the Greek Seas were retrieved from Greek newspapers during a century (1883–1983). Based on the available archives, the first record mentioned the presence of a large 2 m shark in Ionian Sea in 1912 (Table A1, Figure 1a). Earlier from that record, two reports from the literature reported that a shark attacked a sailor in Corfu Island in 1847 [21], and that the bycatch of one specimen of *Carcharhinus* spp. of 4.5 m length occurred in Nisiros Island in 1905 (Dodecanese) (MECO database). Later on, the detection of sharks due to local media, journalists, photographs, better information travel, etc., gradually increased up to 1969, with most records being more frequent for the Aegean Sea (83.0%), whereas the number of records declined over the years leading up to the early 1980s (Table A1, Figure 1a). The spatial extent of the sharks’ presence showed that a quarter of the shark sightings (25.7%) were mostly gathered near of the capital of Athens (Saronikos and Evoikos Gulfs). A significant part was also reported in enclosed gulfs adjacent to large ports, and mainly in cities on the mainland (Thessaloniki and Kavala Bays: 18.1%, Patras Bay: 4.1%) and on the big islands of the North and South Aegean with high touristic activities (Lesvos and Rhodes Islands, respectively) (Figure 1a).

According to the taxonomic resolution of the shark reports, limited reports on this are available, either because during that time the knowledge on these species was in “early stages” or due to the fact that most of the incidents/catches were reported by non-scientific personnel. Thus, apart from many reports mentioning the presence of dog sharks, in very few the presence of *Hexanchus griseus* (three reports) and *Squalus acanthias* (one report; reported as *Acanthias* spp.) was mentioned (Table A1). With respect to the size of the sharks caught, the length was reported in a third of the total reports (33.0%). More than 70% of the sharks caught exhibited, according to the reports, a total length between 2 and 5 m, whereas in exceptional cases (7.0%) sharks of sizes greater than 8 m were also caught.

Nearly 80% of the reports of shark sightings (80.7%) mentioned either the appearance of a shark observed from the beach by a fishing vessel (43.3%) or as the bycatch by the fishery (37.4%), whereas only very small proportion of these reports (5.3%) spoke of the deliberate killing of sharks outside the context of the fishery (14.0%) (Figure 1b). Compared to fishing or shark’s sightings, attacks on people by sharks were spatially restricted to the vicinity of large cities and the coastline (Figure 1b). Only three confirmed lethal attacks of sharks on people were reported between 1900 and 1983, whereas other minor shark attacks occurred mostly due to fishing (Table A1). Attacks on people were mostly reported in the Ionian and Central Aegean Seas, whereas the deliberate killing of sharks were largely reported in the North Aegean Sea (Figure 2a). More than half of the reported attacks on people (58.3%) occurred between 1951 and 1970, when 80% of the deliberate killings on sharks were reported (Figure 2c). Since there have been more shark attacks since 1959, there has been a call for shark culling in collaboration with the appropriate authorities, as well as requests for permission to use weapons to hunt sharks, rewards for those that are killed, and safety swimming recommendations (Figure 3). This was also marked by a gradually increase in the number of shark culling events that were reported during 1959–1967 (Table A1). In certain reports, it was evident that, despite being culled by the coast guard, many sharks belonged to species considered as harmless. Reports also included guidance about safety measures to be taken by sea-users. Moreover, in order to stop shark attacks, various protective measures have been suggested, such as putting special barbed wires in the sea and far enough from the beach (Table A1).

In a third of the reports, the length of shark species was also recorded, indicating that the smallest (<2 m long) and largest (>7 m long) animals were mostly (contribution of more than 40% in each area) reported both in the Evvoikos–Saronikos gulfs and in the Central Aegean Sea (areas with Nos 4 and 6 in Figure 1) (Figure 2c). The largest animals were reduced by 75% during 1951–1960 and 1961–1971, and they completely vanished after 1981, according to the recorded length (Figure 2d). Cases of pregnant female sharks and of premature birth were also reported in certain cases of the sharks caught. However, during that time, there was neither any knowledge on premature care nor on post-release techniques of the embryos, and thus we assume that many embryos were killed. A significant report of a high number of sharks was also recorded after the great earthquake in Santorini Island (9 July 1956) (Table A1).

## 4. Discussion

Historical accounts may be extremely useful to add new data to past shark occurrence and distribution, especially in poorly studied regions such as the waters of the Eastern Mediterranean [7,22]. This allow us to better understand temporal changes in species diversity and population trends, in a location where vast declines and local extinctions are taking place [16,23,24]. In our work, shark reports were spatially extended in most Greek waters, as is the case nowadays [18]. Our historical search showed that about 60 years ago, human beings and sharks were still competing for the same space and large shark species were common in the Greek Seas, a phenomenon not suspected from datasets of modern fishery surveys. However, in several locations, such as the South Aegean waters (i.e., Cyclades, Cretan Sea), there are limited references to shark sightings, especially after 1940. Although there are still significant gaps in knowledge in several Greek areas, such as the Cretan Sea [18], we consider that this issue is mostly attributed to a lack of information or due to the absence of local archives from these areas. Therefore, our work cannot be used for estimating species distribution and abundance during the studied period. The same is also true regarding the trend of the length classes reported across years. Even though there is evidence of a decrease in the length of sharks caught over time, particularly after 1970, this could not be confirmed by our findings because either the reported animals were not disaggregated by species and belonged to a variety of species with different sizes, or the length measurements were frequently based solely on personal observation rather than any scientific methodology.

In the past, some taxa, such as elasmobranchs, were richer in species and more abundant [25]. Currently, shark attacks incidents do not exist in Greece, but this was not true during the studied period. It is well known that the Mediterranean Sea is a hotspot for shark extinction globally [26,27], primarily as a result of overfishing [15,24], but also due to habitat degradation [15]. Although [28] underlines a direct link between shark attacks and the increasing density of sea-users, we must further consider the possibility that some shark species have shifted their distribution as a result of the increasing human pressure on the coastal ecosystems and urbanization (increase in population in the two biggest cities in Greece by more than 130% between 1951–2021: [29]). Such changes have been described elsewhere [23,30], while the recent continuous observations of blue shark sightings in ports and beaches in the Mediterranean (including Greece) during the COVID lockdowns (pers. Comm. with the MECO projects Spain, France, Italy, Croatia, Albania, Greece and Turkey) further support this theory.

Only three lethal attacks of sharks on people in Greek waters during the 20th century were confirmed by our study (i.e., in 1948 at Keratsini area, in 1951 at Corfu Island, and in 1963 at Pagasitikos Gulf), which were also included in the ISAF database. In this database, a couple of other unidentified cases of lethal attacks of sharks on people were also reported, but they were not supported by any publishing reports on the newspapers and thus we consider them as false cases.

Interestingly, the historical records showed that the presence of sharks was a significant issue for coastal communities, and the solutions suggested in the past were related to intentional killings, which are now largely regarded as socially unacceptable and, for some shark species, even illegal. These events were calling for the culling of sharks in co-operation with the competent authorities, requesting the permission to hunt sharks with firearms and offering rewards for killed animals. This was mostly evident in the North Aegean Sea, where local fishers’ associations were leading the efforts of the deliberate killings of sharks and of the free use of guns (see also Table A1 and Figure 2a), whereas in other areas, the co-operation was absent, and all the caught specimens were the work of single fishers. In Greece, the first national legislation protecting two shark species (PD 67/1981) arrived 40 years ago. In contrast, shark-related policies did not arrive in the region until the 1990s, and even now, sharks are still treated and managed as fish stocks rather than as biodiversity. Given that their attitudes and perceptions were formed during the time when the intentional killing of sharks was a reality, citizens and fishers still fear sharks and perceive them as enemies and competitors. Although perceptions against sharks are shifting [5,31], their protection, which is supported by a variety of legal provisions, both national and European, is far from being taken for granted, both because of the delay in time by which these laws are enacted and/or ratified, as well as the lack of enforcement and compliance by professionals. This “back to the history” journey demonstrated that conservation requires patience and persistence, which will enable us to create eco-friendly conservation tactics and policies.

Last but not least, misidentifications, the dissemination of false information, depictions of sharks as prizes, and fear-inducing stories are still widely utilized in Greek media, illustrating the lack of understanding and misinformation about sharks [32]. For conservation priority of species such as sharks, where press coverage has the ability to influence public opinion, correct information is especially crucial; otherwise, the public will continue to associate sharks with negative traits that will hinder any conservation effort [33]. After all these decades of misinformation and fear around sharks in Greece, researchers and conservationists have to invest in training journalists in national and local media, as this can have an immediate impact on the species conservation in the country.

## 5. Conclusions

Our historical search revealed 197 historical records on shark sightings in the Greek Seas retrieved from Greek newspapers between 1883–1983. Such efforts are of particular importance, especially for understudied areas such as the Eastern Mediterranean where information about elasmobranch is lacking. Based on our results, we suspect that shark population in the past were either more abundant or distributed closer to the coast and further research is required to confirm or to reject these theories.

Shark conservation and policy-related instruments are still “young” and the endeavor needs to continue and be intensified in the future, given the gravel situation of their population both globally and in the Mediterranean. In this context, professionals involved in shark conservation need to focus more on the low compliance with current laws, which may be caused by the attitude that people developed when they were young that sharks are competitors and enemies. Based on our search and other available bibliography for Greece, misinformation about sharks has been a persistent phenomenon for over a century, shaping negative attitudes about sharks in the country and jeopardizing any conservation effort. Environmental journalists can serve as a link between people and governments, as well as between governmental and non-governmental organizations while tackling misinformation.

## Figures and Tables

**Figure 1 animals-12-03575-f001:**
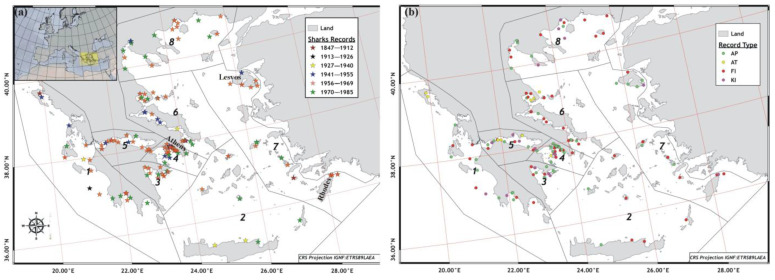
References on presence of sharks during the early fisheries period of the Greek fisheries in terms of: (**a**) year period and (**b**) type of reporting. 1: Ionian, 2: South Aegean–Cretan Seas, 3: Argolikos Gulf, 4: Evvoikos–Saronikos gulfs, 5: Gulf of Corinth, 6: Central Aegean Sea, 7: Eastern Aegean Sea, 8: North Aegean Sea. Record type: AP, appearance; AT, attack; FI, fishery; and KI, killing.

**Figure 2 animals-12-03575-f002:**
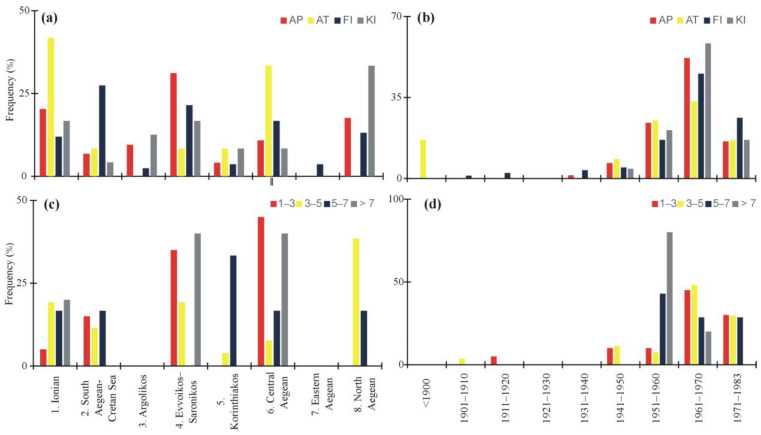
Percentage contribution (%) of shark reports per area and year period in terms of: (**a**,**b**) type of reporting and (**c**,**d**) reported length class (in m). Record type: AP, appearance; AT, attack; FI, fishery; and KI, deliberate killing.

**Figure 3 animals-12-03575-f003:**
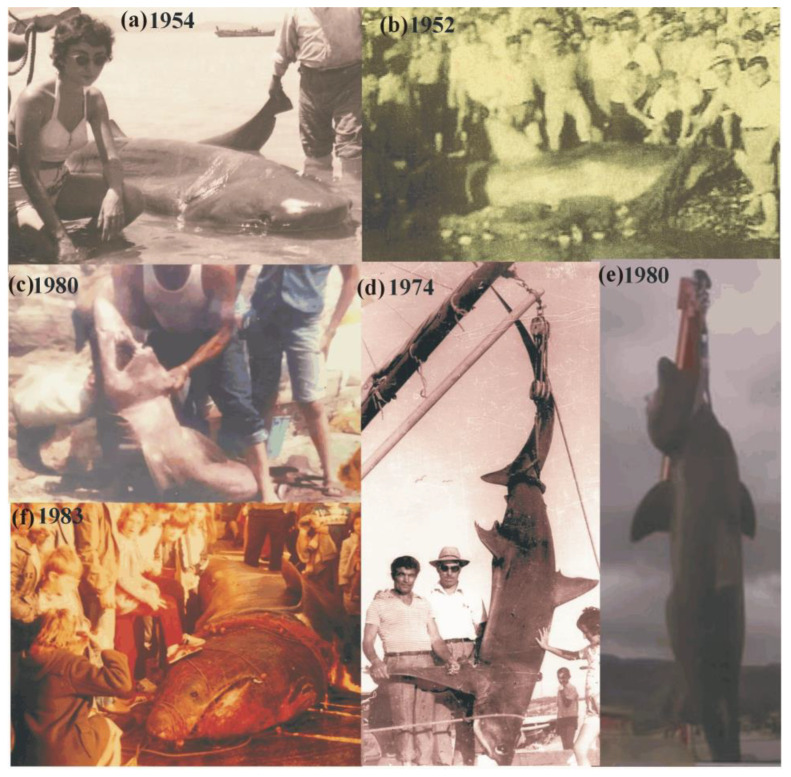
References on presence of sharks during the early fisheries period of the Greek fisheries: (**a**) *Hexanchus griseus* caught in Evvoikos gulf, (**b**) *Carcharodon carcharias* caught in North Aegean Sea, (**c**) Odontaspis ferox caught in Eastern Aegean Sea, (**d**) *Alopias superciliosus* caught in Argolikos gulf, (**e**) *Cetorhinus maximus* caught in Central Aegean Sea, (**f**) *Cetorhinus maximus* caught in North Aegean Sea.

## Data Availability

Not applicable.

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
