# Peer review of "From Extermination to Conservation: Historical Records of Shark Presence during the Early and Development Phase of the Greek Fishery"

_animals, 2022, doi:10.3390/ani12243575_

Round 1

Reviewer 1 Report

Excellent paper, slight revisions required only:

- English is good, but requires some polishing, please check throughout the manuscript.(Example: large shark of 2 m would be better as a large 2 m shark, line 107 limited reported is incorrect, it should be changed to something like limited reports on this are available, etc).

- Abstract - "The key-note fact is that a high number of these observations call from shark attacks on humans, whereas this is no currently evident, resulted as an indicator of lower abundance in modern times." - Not only lower abundance, but changes in water use, as well as more data on shark attack periods and informed bathers… please alter.

- line 93 - "presence of sharks was gradually increased up to 1969 with most records"  - the presence of sharks may or may not have increased, but detection due to local media, journalists, photographs, better information travel, etc was certainly higher… thus resulting in higher shark DETECTION. Please alter.

- Figure 1 (a) and (b) both requires a north. 

- Figure 2 is great, but would benefit from cropping the greek text below the photos. Also, what are (a) and (b) in the legend? Also, insert the year and possible species for each photo in the legend.

- Line 155 - Correct Elasmobranches to Elasmobranchs.

- Line 175 - 2 shark species should be two shark species (number from 1 to 9 are written in full)

Author Response

We would like to thank the reviewer 1 for their constructive comments and suggestions. These were used to revise and improve the manuscript animals-2069756 and also to clarify certain aspects that may have not been previously explained adequately. In the attached file we provide detailed responses to the comments, which were all included in the revised MS.

Reviewer 2 Report

The study has been taken difficult task to combine qualitative information from quite long period in East Mediterranean Sea. Congratulations.

I would like to recommend some small changes in the paper;

1- line 60-61 needs a references. 

2- Please mention about map drawing program in the Material Method

3-Please, mention about percentage calculation for presences of sharks in the Material Method. 

4- Line 210, the sentence which is mention about targeting journalist, please explain well relation journalist and conservationist. You can add one more sentence to attribute potential benefits of cooperation with journalist and conservationist.

Author Response

We would like to thank the reviewer 2 for their constructive comments and suggestions. These were used to revise and improve the manuscript animals-2069756 and also to clarify certain aspects that may have not been previously explained adequately. In the attached file we provide detailed responses to the comments, which were all included in the revised MS.

Reviewer 3 Report

The authors propose an article based on the valorisation of historical data of shark sightings in Greek waters going back to the beginning of the 20th century. The data collected is invaluable and undeniably deserves to be published. However, publication as is seems impossible from my perspective given - as major caveats:

-       The quality of English which is so poor that the very understanding of some ideas is impaired; I have refrained from mentioning all the turns of phrase that make reading laborious. Authors should definitely get help in this aspect of correct expression in English.

-       the caveats linked to the analysis and presentation of the data. While figures 1 and 2 deserve to be maintained, table A1 as it stands is unacceptable (I develop this point below).

The introduction must be improved at least in terms of form (editing of English) and the authors must show more restraint in expressing the quality of the work they believe they have done, as well as with regard to certain databases on shark attacks (in this respect it would be interesting to include in their approach the verification that the greek cases of shark bites on humans are well recorded in the ISAF ? which according to them is exhaustive...).

The results section should give the reader a better overall picture of the most frequent species, special cases, distribution between species and areas, etc. instead of forcing the reader to go to the otherwise very badly presented Table A1. This exhaustive table should be divided into two (or even three) specific tables: first and foremost i) the species identified, ii) the observation conditions (understanding this aspect via Figure 1b is extremely laborious and off-putting). Probably a histogram of the observations by year, differentiating the conditions (AT, FI and AP) would also be useful for an overall assessment, which is not possible with table A1, which can only be kept in the appendix. It should be noted that this table as it stands is inadmissible because of a deficient translation. The information must be both simplified and standardised in a new version, with sub-tables that should be included in the body of the text (as suggested above). An effort to better synthesise and analyse the raw data is needed before publication. Re-conditioning of the data could even help to identify trends in the figures that the authors may have missed… An effort is also needed to extrapolate the species of sharks, either by size or weight when possible, that are mentioned under the simple name of "shark". I find it hard to believe that scientists working in their own country with living fishermen cannot better identify the species that were observed in those days.

A further analytical effort could perhaps improve the discussion on form but also on substance. It is legitimate to include in this discussion the themes of attacks, the global depletion of sharks and the persistent inadequate perception of these animals in Greece.

The following table specifies specific problems to be solved in the new manuscript. However, this approach cannot avoid a total revision of the manuscript by taking up the analytical approach to content and form.

Line

Comment

3 -Title

The fact that these records are “unpublished” is obvious and I guess the word “historical” might be more relevant

12

“exhaustive”: There is no evidence that the research was totally exhaustive as the authors assume, so the term 'thorough' would seem more appropriate

16-17

“In the 16 early 20th century (1913) the first record on the presence of a large shark of 20 m of length reported 17 in the North Ionian Sea”: the work “was” is lacking before reported ?

17

“20 m” : what species was it ? isn’it rather “2 m” ?

18

“Later on, the presence of sharks was gradually increased up to 1969”: the “was” needs to be removed

20-21

“The key-note fact is that a high number of these observations call 20 from shark attacks on humans, whereas this is no currently evident, resulted as an indicator of lower 21 abundance in modern times.” I don't understand this sentence.

25-37

I don't see much difference between these two abstracts ? neither in length nor contents: do they really fit with the guidelines of the journal ?

59-63

“is the world’s only scientifically 60 documented, comprehensive database of all known shark attacks. ». This is not true as the Global International Shark Attack files is also fully reliable. And ISAF is n ot as exhaustive as it should be. A scientific paper  is not a vulgar propaganda tool, let alone a misleading one. A certain restraint is expected from the authors and is an integral part of the basic deontology.

97

The capital is not shown on the Fig. 1a. It should be. If not, what interest to cite it ?

100-101

Same comment as above for Lesvos and Rodhes islands, etc.

107-109

very confusing expression to simply say that the taxonimic resolution is low

12!-129

“In certain reports, it was evident that despite the killing of sharks by coast guard authorities, the latter were considered as un-harmed these species ». Difficult to understand ? Do the authors mean that “Despite being culled by coast-guards many sharks belonged to species considered as harmless ?”

140-141

The caption of the figure is probably wrong. A suitable caption should indicate details about what is shown on the photo, in particular which shark species and which size ? as well as the location of the photo

130-131

“In the reports there were also guides to people being aware in the beaches as well as to professional and recreational fishers.” Being aware about what ? Do the authors mean that “Reports also included guidance about safety measures to be taken by sea-users” ?

159-160

“Increasing shark attacks cannot be directly linked with larger populations [21], but in the Mediterranean context might be a plausible explanation. » “Larger populations” of what ? sharks ? people ? I don't understand this sentence and I am not sure that the authors have fully understood the conclusion of West (2011) that underlight a direct link between shark attacks and the increasing density of sea-users.

164-167

I don't understand the demonstration that links a paper (Hammerschlag et al. 2022) showing that shark densities are (unexpectedly) high in anthropogenized spots and the COVID effects that have allowed sharks to get observed in  beaches where the human density has dropped ? the reasoning undoubtedly needs to be better structured and clarified

Author Response

We would like to thank the reviewer 3 for their constructive comments and suggestions. These were used to revise and improve the manuscript animals-2069756 and also to clarify certain aspects that may have not been previously explained adequately. In the attached file we provide detailed responses to the comments, which were all included in the revised MS.

Round 2

Reviewer 3 Report

Although it has been improved and is understandable, this article still needs a better english editing.

At least the abstract should be corrected as follows:

The lack of historical data on shark presence, distribution, and status in the Eastern Mediterranean undermines efforts to manage and protect their populations. An exhaustive review of anecdotal references related to shark presence during the early and development phase of Greek fisheries (1883-1983) was conducted. In the early 20th century (1913) the first sighting on the presence of a dead shark was reported in the Ionian Sea. Later on, the presence of sharks gradually increased up to 1969 with most records being more frequent for the Aegean Sea, whereas the number of shark presence reduced up to the middle of 1980s. The increase of shark attacks during the mid-20th century was calling for culling of sharks in co-operation with the competent authorities promoting the permission to hunt sharks with firearms and offering rewards for killed individuals. A high number of these observations potentially call from shark attacks on humans, whereas this is no currently evident, resulted as an indicator of lower abundance in modern times and subsequently an alteration in the way that our current modern society is approaching the protection of such vulnerable species. 

 [EC1]Not clear ; should be split into two sentences and clarified

Author Response

We would like to thank the reviewer 3 for their constructive comments and suggestions. These were used to revise and improve the manuscript animals-2069756 and also to clarify certain aspects that may have not been previously explained adequately. In the attached file we provide detailed responses to the comments, which were all included in the revised MS.

Yours sincerely,

Associate Professor Dr. Dimitrios K. Moutopoulos
